# Optical Transformers

## Abstract

The rapidly increasing size of deep-learning models has caused renewed and grow-
ing interest in alternatives to digital computers to dramatically reduce the energy
cost of running state-of-the-art neural networks. Optical matrix-vector multipliers
are best suited to performing computations with very large operands, which leads
us to hypothesize that large Transformer models might achieve asymptotic energy
advantages with optics over running digitally. To test this idea, we performed
small-scale optical experiments with a prototype accelerator to demonstrate that
Transformer operations can run on optical hardware despite noise and errors. Using
experiment-calibrated simulations of our hardware, we studied the behavior of
running Transformers optically, identifying scaling laws for model performance
with respect to optical energy usage and estimating total system power consump-
tion. We found that the optical energy per multiply-accumulate (MAC) scales as
$\frac{1}{d}$ where $d$ is the Transformer width, an asymptotic advantage over digital sys-
tems. Should well-engineered, large-scale optical hardware be developed, it might
achieve a $100\times$ energy-efficiency advantage for running some of the largest current
Transformer models, and if both the models and the optical hardware are scaled
to the quadrillion-parameter regime, optical computers could have a $> 8,000\times$
energy-efficiency advantage over state-of-the-art digital-electronic processors (300
fJ/MAC). We discussed how these results motivate and inform the construction of
future optical accelerators and optics-amenable deep-learning approaches. With
assumptions about future improvements to electronics and Transformer quantiza-
tion techniques (5× cheaper memory access, double the digital–analog conversion
efficiency, and 4-bit precision), we estimated that optical computers' advantage
against these digital processors could grow to $> 100,000\times$.

## 1   Introduction

Deep learning models' exponentially increasing scale is both a key driver in advancing the state-of-
the-art and a cause of growing concern about their energy usage, speed, and practicality. This has led
to the development of hardware accelerators and model training/compression/design techniques for
efficient and fast inference on them.

While digital-electronic accelerators [47, 16, 8, 1, 17] can improve performance by some constant
factor, alternative analog computing platforms using optics have been proposed as a new paradigm
for better scalability [49, 7, 62, 41, 56, 24, 51]. Ideally, the scaling is asymptotically better than
digital systems in energy per MAC [18, 61, 53, 41]. But these optical neural networks (ONNs) have
additional complexities and limitations of their own such as low precision, noise, and analog/digital
data conversion overheads which depend on the access patterns of the model running (Figure 1).
Thus, advantageously accelerating any neural network architecture with ONNs is hard. Here, we
hope to answer whether Transformers' efficient data-access patterns (wide layers, parallel/batched

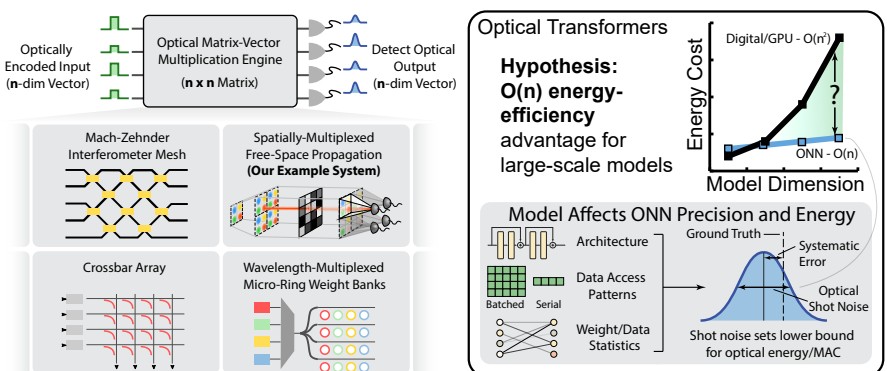

Figure 1: **Can Transformers Benefit From Running on Optical Hardware?** Optical Neural Networks (ONNs) have been proposed as an alternative computing platform that can achieve asymptotic energy-efficiency advantages over digital computers running neural networks. This is not a guarantee; their behavior is affected by model architecture, statistics, and resilience to the noise/imprecision of analog hardware. Thus, while there are many implementations of general-purpose optical matrix accelerators (such as those depicted in the inset), there are still model-dependent challenges/tradeoffs in realizing their purported advantages. We seek here to answer the question of how much today's enormous Transformer models can benefit from this technology, if at all. Our hypothesis is that Transformers' architecture and unique behaviors allow for ONN-enabled benefits that scale.

token processing, etc.), trends in methods for scaling them, and sufficient effort to train them for ONNs afford them the asymptotic energy-efficiency advantages of running optically.

Here we demonstrate how the popular Transformer architecture is able to run on ONN systems, and estimate the potential benefits of doing so. To first verify that Transformers may run on these systems despite their imprecision, we sampled operations from a Transformer and ran them on a real spatial light modulator (SLM) based experimental system, and used the results to create a calibrated simulation of the optical hardware, with the systematic error, noise, and imprecision of weights/inputs we observed. Transformers running on the simulated hardware could perform nearly as well as those running digitally, and could be far more efficient. We summarize our key contributions as follows:

- We demonstrated linear Transformer operations (the bulk of a Transformer's computation) running with sufficient accuracy on real optical hardware and in a matching simulation, despite errors and noise.
- Via simulation, we established scaling laws for optical Transformer performance versus optical energy usage, and optical energy usage versus model size.
- Based on our simulations and experiments we estimated an orders-of-magnitude energy consumption advantage of full ONN accelerators versus state-of-the-art GPUs.
- We discussed Transformers' suitability for optical acceleration, and more generally how specific elements of DNN architecture affect the function of ONN systems running them.
- We identified the hardware and systems design challenges that future work on building ONN accelerators should target.

While our experiments and simulations were based on specific hardware as a representative example, our scope here is more general. We are interested in understanding how uniquely optical energy scaling and noise relate to Transformer performance and architecture. As such nearly all our findings apply broadly to linear optical processors (and hopefully future ones), irrespective of their underlying hardware implementation details.

## 2 Background and Related Work

### 2.1 Transformer Models

Transformers are models for processing sequential data based on multi-head attention. Transformers consist of two-layer feed-forward blocks and multi-head attention (Figure 2) operations. Multi-

head attention computes relationships between sequence elements by deriving query, key, and value sequences $Q, K, V$ and computing dot products with a softmax nonlinearity in-between [60]. Transformers also leverage modern design elements such as additive residual skip connections [20] and normalization layers [3]. A defining feature of Transformers is that entire sequences may be processed in matrix-matrix products in parallel (instead of one token/input at a time).

## 2.2 Large-Scale Deep Learning

In the past few years, it has been found in particular that Transformer [60] architectures significantly improve when sized up to billions or even trillions of parameters [6, 28, 10, 22, 59, 66], causing an exponential growth of deep learning compute usage [48, 50]. These large-scale Transformers achieve ever more impressive results in not only natural language processing, but also in other domains such as computer vision [14, 36], graphs [30], and in multi-modal settings [27, 26, 44, 45, 65, 46], making them a popular but expensive solution for many tasks—digital hardware's energy efficiency (ie. per-flop or per-inference cost) has not kept up with the growing FLOP requirements of state-of-the-art deep learning models [50]. They also have transfer learning capabilities [42, 13, 43, 6, 37, 14], allowing them to easily generalize to specific tasks, in some cases in a zero-shot setting where no further training is necessary [6, 45, 33].

## 2.3 Optical Accelerators

Researchers have explored a wide variety of controllable optical systems which manipulate different types of optical modes to effectively implement arbitrary matrix-vector multiplications, vector-vector dot products [52, 2, 18, 55, 4, 61, 19, 39, 57], or convolutions [63, 15, 40, 64]. In this work, we adopt the free-space multiplier [61, 55, 19] (Figure 2, top left) to demonstrate Transformer operations in optical experiments and for our simulations. We selected this system because it has many of the same behaviors as other ONN implementations, and aim to draw conclusions that could generally be useful for those working with other ONN designs. Many ONN systems, including ours, share the following typical traits:

**Device Imprecision and Optical Shot Noise**   Optical systems are subject to errors in both the actual hardware and from photon detection. Detection of optical intensity in particular is subject to a phenomenon known as *shot noise* where the detected value is Poisson distributed: given vectors $x$ and $w$, with the elements of $x$ encoded as optical intensity, the output $Y$ is distributed as:

$$Y \sim \text{Poisson}(w \cdot x) \tag{1}$$

For other encoding schemes such as amplitude or phase encoding, equation 1 should be modified, but the detection is still subject to shot noise.

**Efficient Photon Usage**   Shot noise, and therefore an optical dot product's signal-to-noise ratio (SNR, which serves as an effective bit precision) is related to the mean number of photons at the *output*. The efficiency of photon usage can therefore grow with increasing multiply-accumulate operations (MACs): the SNR for the product $w \cdot x$ is

$$\text{SNR}(Y) = \frac{\text{E}[Y]}{\sqrt{\text{Var}[Y]}} = \sqrt{w \cdot x} = \sqrt{\text{E}[Y]}, \tag{2}$$

which explains this behavior; if the desired output precision does not change, constant photons are required regardless of dot product size. Work on ONNs has studied this behavior in a variety of scenarios [18, 41, 61, 53]. This efficient scaling is not a guarantee—the required number of photons may be influenced by a model architecture's activation/weight distributions, encoding schemes, precision requirements, etc.

**Optical Neural Network Energy Costs**   The energy cost of optical neural networks is broken down into the optical costs of performing MACs and the electrical costs of loading/detecting data, which are usually dominant. Consider a product between two matrices, $A \in \mathbb{R}^{n \times d}$, $B \in \mathbb{R}^{d \times k}$. Such a product results in loading (detecting) $nd + dk$ ($nk$) scalars, and performing $ndk$ MACs. If the energy

to electrically load (detect) a scalar is $E_{\mathrm{load}}$ ($E_{\mathrm{det}}$), and to perform a MAC optically is $E_{\mathrm{optical}}$, then the total energy is:

$$E = (nd + dk)E_{\mathrm{load}} + nkE_{\mathrm{det}} + ndkE_{\mathrm{optical}} \qquad (3)$$

This illustrates how ONNs may have asymptotic energy advantages over digital computers. Notice that regardless of the number of reuses, all data is only loaded once in Equation 3. This is because copying a vector's data and transporting it is free optically. Meanwhile, $E_{\mathrm{optical}}$ ideally scales as $1/d$. These properties make energy cost disproportional to the number of MACs, $ndk$. In other words, $\frac{E_{\mathrm{digital}}}{E_{\mathrm{ONN}}} \sim \min(n, d)$.

**Streaming Weights Versus Weights-In-Place** There are two approaches for loading weights. *Weights-in-place* schemes involve loading them once, and re-using them for many inputs. Alternatively, systems can employ *streaming weights* where at every computation the required weight matrix is loaded. Our experimental system is a weights-in-place scheme. For weights-in-place operations, the energy advantage scales as just $\frac{E_{\mathrm{digital}}}{E_{\mathrm{ONN}}} \sim d$.

## 2.4 Previous Optical Neural Network Architectures

Previous work has considered deep learning models such as MLPs and convolutional networks on benchmark tasks like MNIST [40, 61], and simulations of larger convolutional models such as AlexNet [32] on more difficult datasets such as ImageNet [18]. This begs the question of how well newer, larger models perform on optical systems.

## 2.5 Scalable Compression and Quantization of Large Language Models (LLMs)

Optical hardware's low precision raises the question of whether scaled-up models could be quantized sufficiently to run. Thankfully, continual research in LLM compression has progressively shown that larger models do not have increasing precision requirements. For example, [34] found that larger Transformers can be compressed more easily, to the degree that it is more worthwhile to train large ones and compress them over training smaller ones of the target size. Furthermore, [5] and [12] demonstrated running Transformers at scale with int8 precision, and the recent work of [11] proposes that 4-bit is optimal for nearly all model scales, except for the largest tested (175B parameters) where 3-bit was sometimes found to work better.

# 3 Optical Transformers

We designed models that are intentionally similar to other Transformers, with the goal of simulating their behavior (informed by some experimental measurements) and energy consumption on optical hardware. A summary of our approach and model is in Figure 2.

## 3.1 Architecture and Task

We created optical Transformer models with a GPT2-like [43] architecture that replaces the GELU [21] activation with ReLU6, which is known to improve low-precision model performance [31, 23, 29]. For language modelling, we used the raw Wikitext-103 dataset [38]. The models we simulated have 12 layers (consisting of multi-head attention and feed-forward blocks), operate on a context length of 1024 tokens, use 12 attention heads, and have embedding dimension $d$ varying from 192 to 1536. The full details of the training technique, architecture, and hyperparameters are in Appendix A.

## 3.2 Transformer Computations on Optical Hardware

We ran experiments using a real Transformer's (we used the base-sized model with $d = 768$) weights in order to characterize the behavior of an ONN system. We adopted as a representative example of an optical accelerator a spatial light modulator (SLM) based system which computes vector-vector dot products [61]. Vectors are encoded on a display, and copies are shone through the SLM which has varying transmission corresponding to some data (ie. a weight matrix). The outputs of this

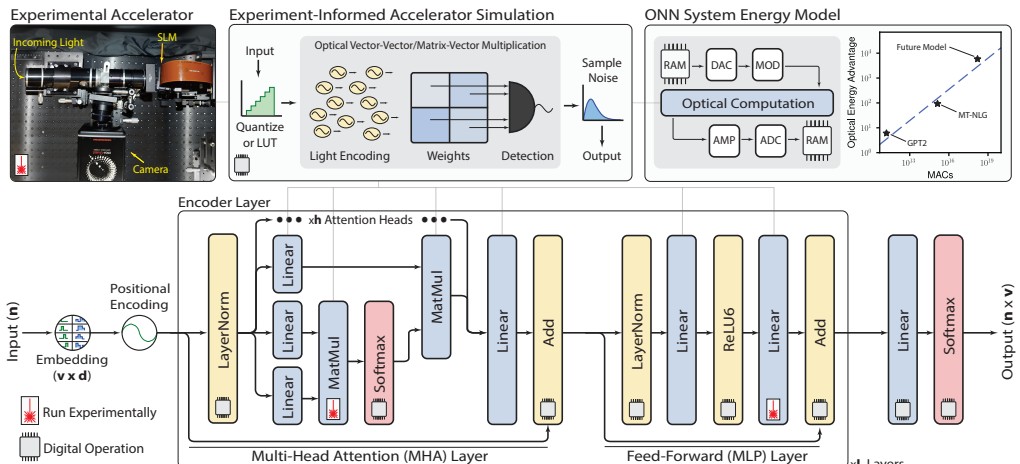

Figure 2: **Optical Transformer evaluation: prototype hardware; simulator model; Transformer architecture.** Bottom: typical Transformer architecture, but with ReLU6 activation. Top Left: experimental spatial light modulator (SLM)-based accelerator setup. From some layers—marked with a laser icon—we sampled dot products to run on real hardware. Top Middle: Linear operations, in light blue, run on a simulated accelerator with noise/error. Lookup tables (LUT) allow simulation using our setup's supported weight/activation values. Top right: our model of energy consumption for optical accelerators, based on assumptions and results from our experiment/simulations. The model accelerator system consists of random-access memory (RAM), a analog/digital conversion (DAC/ADC), light modulation (MOD), amplification (AMP).

operation—element-wise products—are collected at detectors as the resultant dot products (Figure 2, top left). We collected lookup tables (LUTs)—mappings of the available discrete levels in both the display and SLM devices—and used them to train a "LUT-aware" optical Transformer model to run on the setup. We then collected calibration curves, mappings from the detected output light intensity to the actual neuron floating-point values. To do this, we ran many random dot products on the hardware and collected pairs of detected values and digitally-computed ground-truth values. We then fit the relationship linearly. We used high photon counts to eliminate shot noise, so deviation from the linear fit was considered the hardware's *systematic error*. Full details of experimental procedures and calibration are in Appendix B.

## 3.3 Simulation of Optical Hardware

Informed by our experiments, we constructed a simulation of the optical hardware. By simulating the hardware behavior directly we model how any arbitrary operation would behave if run on the physical setup. This allows us to avoid the computationally demanding task of simulating much larger Transformers to verify that our simulation method works. We aimed to emulate the noise, error, and preci-

Table 1: Summary of simulation configurations for different evaluation and training scenarios. For simulating optical hardware we included all behaviors. For determining optical resource scaling, we focused on shot noise, and ran a plain 8-bit model for comparison.

| Setting | Op. | Shot Noise | Sys. Err. | LUT | 4-Pass |
|---------|-----|-----------|-----------|-----|--------|
| Hardware | QAT | ✗ | ✗ | ✓ | ✗ |
| Simulation | Eval | ✓ | ✓ | ✓ | ✓ |
| Optical | QAT | ✗ | ✗ | ✗ | ✗ |
| Scaling | Eval | ✓ | ✗ | ✗ | ✓ |
| Simulation | Int8 | ✗ | ✗ | ✗ | ✗ |

sion that we observed in order to understand how well full Transformers would perform when running on optical hardware. The configurations for different scenarios are summarized in Table 1. We also evaluated the digital, 8-bit-QAT-trained model for comparison purposes.

**Hybrid Scheme**  Pure optical systems cannot easily compute activation or normalization functions. Thus we assumed LayerNorm, ReLU activations, and residual skip connections are performed digitally at full precision. Thankfully, even in smaller models, linear computations are the overwhelming majority (Section 4.3).

**Non-Negative Weights and Inputs ("4-Pass" Multiplication)**    An important limitation is that our display and SLM only support non-negative values. The constraint of having all-positive data is present in many but not all optical neural network systems.We worked around this by decomposing products into sums/differences of products with non-negative operands. Consider a product between matrices $W$ and $X$. If we let $W_+$ ($X_+$) and $W_-$ ($X_-$) be matrices with only the positive and negative elements of $W$ ($X$) respectively, then:

$$WX = W_+X_+ - |W_-|X_+ - W_+|X_-| + W_-X_-  \qquad (4)$$

**Data Scaling**    On the real system, we define a maximum activation/weight value as 1.0 and minimum as 0.0. To simulate operation, the inputs and weights of every simulated NN layer are scaled to this range, and then rescaled back afterwards.

**Device Quantization**    Real hardware may only have certain number of representable levels. To emulate this behavior, we fine-tuned pretrained models using quantization-aware training [25](QAT) and applied the following in simulation (hyperparameters in Appendix A):

- For optics-simulated layers, we emulated quantization to int8 (256 levels). Then, instead of dequantizing, we used the integer values directly as indices into the LUTs that we gathered from experiment.

- We also quantized weights, but with the SLM LUT. We clamped smaller values to 0.02 in the simulation, as our SLM does not have a high extinction ratio, and the smallest transmission is 0.02.

- Accumulation can be high precision, but we used int8 quantization for outputs, since analog-digital conversion (ADC) is expensive in practice.

- We used both deterministic and stochastic rounding when quantizing, with similar results.

**Systematic Errors**    Issues like cross-talk, misalignment, defects in ONNs give rise to systematic errors. We simulated such a constraint by adding Gaussian noise to simulated model outputs (Figure 2), scaled relative to the mean sizes of the outputs, as this was the noise behavior we observed experimentally (it is related to the rescaling of data between 0 and 1).

**Optical Encoding and Shot Noise**    We modeled optical encoding by subjecting layer outputs to simulated shot noise (Figure 2), which differs from the systematic error model. Outputs were scaled by a number such that the average photon number per feature (photons/MAC) was some target value. Each of these features was used as the mean of a Poisson distribution, which we sampled. These outputs were then scaled back down to represent neuron values. In the simulations for optical scaling we used vanilla 8-bit QAT (no LUTs or systematic error, which can overwhelm shot noise) to cleanly demonstrate the optical scaling properties—which are model-dependent and not hardware-dependent—of Transformers.

## 4   Results

### 4.1   Transformer Error Tolerance and Hardware-Simulation Accuracy

We determined experimentally that Transformer operations are able to run on real hardware without severely degraded performance from systematic errors. The bottom four panels of Figure 3 are histograms of the experimental differences from correct values. The simulated noise distributions (dotted lines) match well with the experimental data, which confirms that they are an accurate representation of the real systematic error behavior. Figure 3 (top) is a map of the performance of the simulated model over different configurations of the mean-relative (in percent) noise at every layer of feed-forward and attention blocks. The model performs well with significant noise (experimental noise levels marked with stars), within 1 perplexity from noise-free performance unless the noise is very high. These results show that our digital model of the system is a plausible approximation of how a real one might behave.

While 8-bit precision was used for QAT, the optical Transformer can perform inference at lower precision, as implied by its error tolerance. To study this further we conducted a simple ablation on the input and output precisions used at inference, on the 8-bit-QAT base-sized model with LUT in Appendix C.

## 4.2 Optical Scaling Laws

Optical Transformers achieve language modelling performance close to their digital counterparts' when shot-noise-limited at modest photon budgets. The perplexities on the Wikitext-103 validation set of various optical Transformer models simulated with different total photon usage (amount used for input data) are shown in Figure 4 (left). The curves illustrate a tradeoff: larger models need larger photon totals to function well, and there are different optimal model choices based on the photon budget. We define photons/MAC as the total photon budget (amount at input) divided by total MACs. The percentage difference from the performance at 10K photons/MAC (Figure 4, middle)—chosen to represent an ideal high-precision scenario—is roughly power-law scaled in photons/MAC for all models with truncation near 10K; better performance can be had with more photons, but with diminishing returns, and the performance matches or exceeds that of the 8-bit digital models' when the photon budget is not too low ($\sim 10^2$).

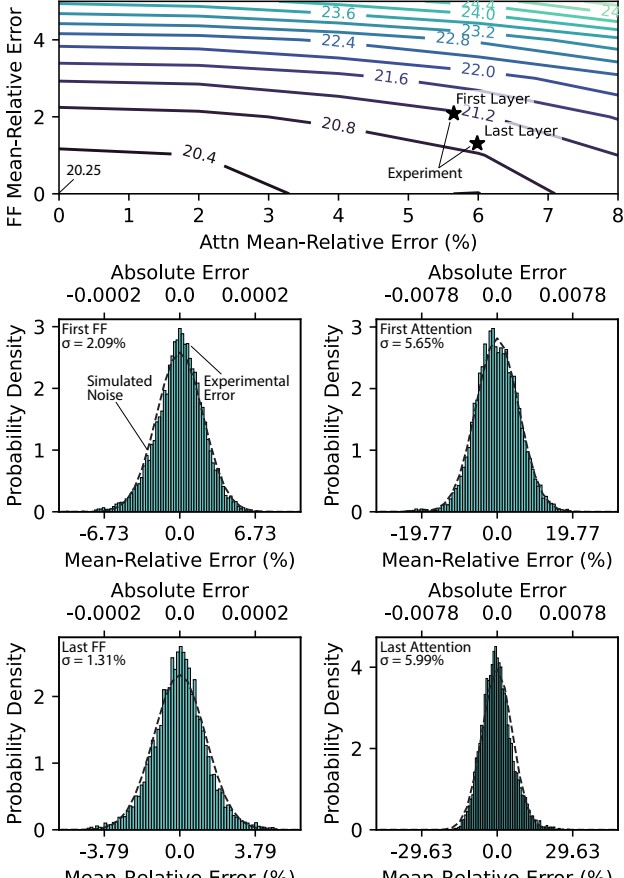

Figure 3: **Comparison of experimental and simulated noise models and simulated Optical Transformer noise tolerance.** Top: Simulated performance (Wikitext-103 validation perplexity (PPL)) versus percent mean-relative simulated noise in feed-forward (FF) and attention (Attn) layers. Systematic errors from experimental data marked with a star. Bottom: comparison of simulated noise model to error from experimental data. The Gaussian shape of the simulated error behavior matches experiment accurately.

The models use fewer photons/MAC as they scale, achieving the theoretical efficient scaling where the total per-dot-product photons needed is constant. To study how photon usage scales, we determined how many photons it takes to reach the performance of 8-bit digital models. These values, in Figure 4 (right), decrease nearly as $\frac{1}{d}$—the total photons needed per dot product is constant (bottom dashed line). The Transformer architecture clearly takes advantage of efficient optical scaling with larger model sizes. In fact, smaller per-dot-product totals are required for the largest model, suggesting that larger Transformers may require less output precision. This is consistent with other work which found that precision requirements are constant or reduced with scale [34]. Meanwhile, the already low photon usage of the largest model suggests that models larger than our simulations (>10B parameters) may use <1 photon/MAC. This sub-photon operation works in optical systems [61, 53] and is in essence no different at all from operation at higher photon counts (since the number summed at detection is still high).

These empirical scaling results are tied to our specific configurations and training strategies. Depending on the scales and dynamic ranges of inputs and weights, different amounts of photons may be transmitted to the output; the statistics of a model affect its efficiency. In Appendix H we explore a different scheme, but the effects of different methods remains an interesting topic for future work.

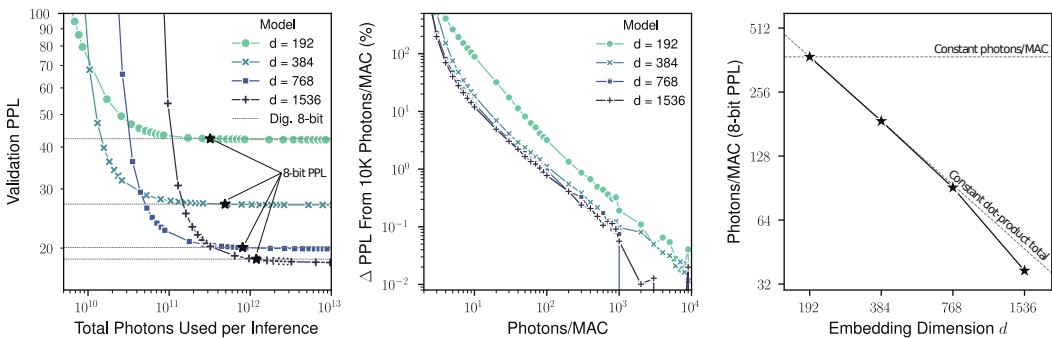

Figure 4: **Simulations of Optical Transformer behavior with varying photon usage.** Left: Wikitext-103 validation-set perplexity (PPL) versus embedding dimension $d$ and total photons used for a single forward pass/inference. 8-bit digital model performance is shown with dashed lines. Middle: perplexity degrades from ideal with fewer photons-per-MAC; the plot exhibits truncated power-law scaling. Right: Scaling of number of photons needed for an Optical Transformer to achieve the same perplexity as an 8-bit digital-electronic processor, versus model size.

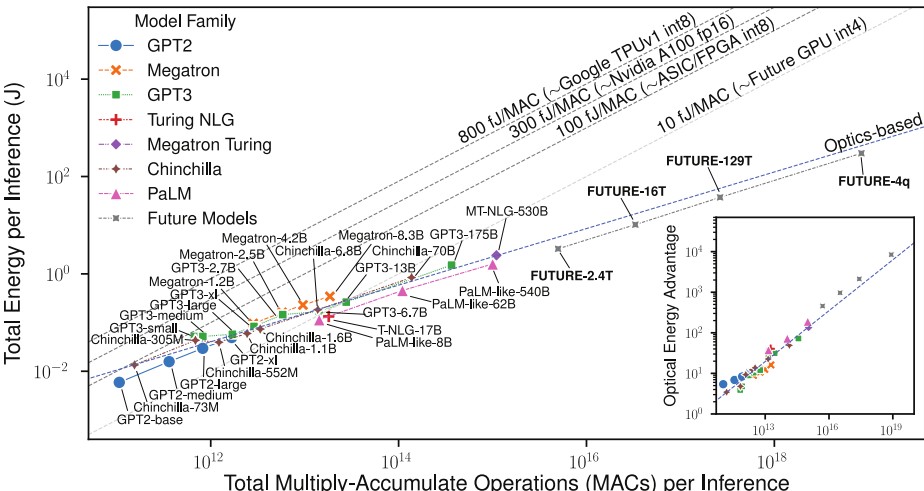

Figure 5: **Estimated energy usage of Transformer models on optical hardware for a single forward pass/inference.** Hypothetical future model designs are labelled **FUTURE-\***. Estimated energy/MAC for digital systems is based on [47]. Trend for energy usage in optical systems (blue) computed based on real models only. Inset: energy advantage of running on optics over estimated NVIDIA A100 usage. The advantage grows with the model compute. $M = 10^6$, $G = 10^9$, $T = 10^{12}$, $q = 10^{15}$ parameters.

## 4.3   Estimated Energy Usage

The efficient photon scaling trend we observed in Section 4.2 suggests that Transformers running on optical hardware could achieve significant energy efficiency advantages over running on digital hardware. To understand the efficiency of Transformers on optical hardware, we designed an ONN system based on current hardware that is like our experimental setup, with our measured precision and photon scaling. It is an inference system with in-place weights which are loaded once and reused forever, activations read from and written to SRAM for every layer, a 10 GHz light modulator array, and an optical "core" which can perform 10M multiplications per cycle (this can be thought of as a 10 megapixel SLM). The photon-per-MAC scaling versus model dimension is taken to be the $1/d$ scaling which we found was possible in our simulations, and we assumed that the model operates with 5-bit input precision, 8-bit weight precision, and 7-bit output precision, as determined by our study of low precision performance in Appendix C. We then calculated according to the approach in Section 2.3. For electrical energy we assumed in-place weights and did not include the energy for loading them. In Appendix D we explain all assumed energy quantities based on contemporary hardware.

As models grow, running Transformers on optical hardware has a large and asymptotic efficiency advantage over running on digital hardware. In Figure 5 we chart estimates of the forward pass energy required for various models[1], including a hypothetical family of large, dense Transformer models designed in a similar fashion, which we label **FUTURE-\***. For comparison, we also chart various digital systems [47] in different performance regimes, and a hypothetical "next generation" GPU that can use $\sim$10 fJ/MAC. For small models, the optics-based system uses about the same energy, but eventually gains an advantage that scales asymptotically with the number of MACs. For the larger models, MT-NLG-530B and FUTURE-4q, the optics-based approach would have $\sim$140$\times$ and $\sim$8500$\times$ energy advantages over the current state-of-the-art GPU (NVIDIA A100) respectively.

The breakdown of compute and energy costs by source is in Appendix E. In summary we found that as models get larger the feed-forward layers require most of the computation, but that the energy of data access in attention is still very expensive due to the many heads. This is because of the parallel operation of the Transformer, where the linear layer weights can be re-used for many tokens at a time (weights-in-place is not possible for attention, and there are $h\, n \times n$ attention maps to store). [2]

## 5   Discussion

The results given in Section 4.3 on optical Transformers' efficiency have implications for the design of future ONN hardware/software systems.

In Appendix G we discuss in detail the specifications for an ONN system to run large Transformers, as a target for future work in their design. In summary, we found: once matrix-matrix product operands exceed $10^4 \times 10^4$ in size the advantage is significant, and therefore a future ONN should implement at least this level of parallelism to achieve $>$100$\times$ efficiency improvements over current state-of-the-art GPUs (NVIDIA A100). Given the assumptions we made about weight-maintenance costs in making our estimates (5.6 µW per weight; see Appendix D), an Optical Transformer would need to operate in the regime where a single matrix-vector multiplication is performed every 0.1 nanoseconds. Current ONN prototypes either operate at low clock rate or at small scale. Thus building a full ONN system that realizes the potential benefit is still an open challenge.

Future improvements in CMOS technology will be greatly beneficial. In Appendix F we estimate that future optics-based systems might achieve energy advantages of $>$100,000$\times$ running models the size of FUTURE-4q (over 300 fJ/MAC).

Our studies on Transformers illustrates more broadly the relationships between model design and ONN efficiency. Transformers sought to make large models run efficiently by exploiting hardware's strengths in performing large, parallel, dense calculations, and improved in this aspect as they scaled. As a consequence, as Transformers continue to be optimized for parallel digital electronic hardware, they will continue to become even more efficient on optical hardware. More generally, architectures that perform more computations per data access (such as those focusing strongly on linear operations [58, 35]) will be most promising for optical implementation.

**Conclusion**   We have demonstrated the ability of Transformer models to run accurately and efficiently on optical hardware through optical experiments and an experiment-informed simulation of the hardware. We examined Transformers' scaling behavior with optics and used our findings to show that optical systems could have a large and asymptotic energy advantage over digital ones that *grows* with the model size. For example, we showed that optical hardware may achieve an over 100$\times$ energy advantage when running the largest Transformer models today ($\sim$500 billion parameters) and that larger, future Transformers ($\sim$4 quadrillion parameters) may be realized with an $>$8000$\times$ optical energy advantage. We believe our findings about the potential energy-efficiency of optical accelerator hardware strongly motivate the development of optical processors for large-scale deep learning with Transformers.

---

[1]The recent PaLM [9] models used a modified architecture. For simpler comparison, we make our estimates using a model with GPT-like architecture but with the PaLM model dimensions, which we call PaLM-Like.

[2]Trends in the design of real models have increasingly favored optics over time. Specifically, attention loads/stores a $n \times n$ attention matrix for each of the $h$ attention heads. Models with more MLP compute per attention head have a larger overall ratio of computation to energy usage; larger $\frac{d}{h}$ is more efficient. The largest GPT2 [43] uses $\frac{d}{h} = 64$; GPT3 [6], 128; MT-NLG-530b [54], 160; and PaLM [9], 384.

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
