# OpenReview forum: "Optical Transformers"
_NeurIPS.cc/2023/Conference — Submitted to NeurIPS 2023_

### Official Review · Reviewer_5GP2 · 2023-06-20

**Soundness:** 3 good
**Presentation:** 3 good
**Contribution:** 1 poor
**Rating:** 4
**Confidence:** 5

**Summary:**

The authors analyze the performance, efficiency, and robustness of free-space optical dot-product engines for Transformer accelerations. Measurement results on an SLM-based optical system are demonstrated on some layers in a GPT-like model. System performance/efficiency are estimated and compared to digital computers. Scaling of optical processors are discussed to show the scalability of optical computing platforms.

**Strengths:**

1. Experimental results on SLM-based free-space optical system has been demonstrated for matrix multiplication.
2. Scalability with future technologies are discussed to show the benefit of optical computing in the future.

**Weaknesses:**

1.	The novelty of the paper raises some concerns as no new hardware design or algorithm innovations have been shown. The SLM-based system and its experimental demonstration are not new. No customized hardware is shown for Transformer. The claimed optical hardware is designed for CNNs/MLPs. On the algorithm part, device quantization, the LUT-based training method, noise analysis, and 4-pass multiplication are standard methods for analog computing. NeurIPS community usually requires certain machine learning contributions. What is the main ML contribution? Probably other venues in the optics community are more suitable for this paper.
2. The demonstrated system is weight-in-place which needs a large number of parallel MVM to amortize the weight programming/encoding cost. However, the dynamic attention operations in Transformer and fully-connected layers usually have low arithmetic intensity, especially GPT-like architecture with KV cache, which cannot provide enough batch dimension to amortize such cost for weight-in-place systems. More justification for the usage of the weight-in-place system needs further discussion. A weight streaming system might be the suitable architecture for Transformer.
3.	In Fig. 2, only a small part of layers in the Transformer block are implemented by optics, while other operations are on all digital computers. In this hybrid case, how large are the efficiency/performance benefits, or is it worthwhile to use optics?
4.	For the noise analysis, only shot noise is emphasized, which is much smaller compared to other variations in the system both on the electrical and optics sides. A simple Gaussian added to the output results might be oversimplified as system error modeling.
5.	In Line 291, the system assumes a 10 GHz light modulator array. If I understand correctly, the spatial light modulator typically has high resolution but very low switching frequency. This 10 GHz modulation speed needs further justification. How fast is the switching freq for weights and input feature maps? The modulation energy cost is based on thin-film lithium niobate modulators, which are fairly large. How many such large modulators are required to modulate a million pixels?
6.	Also, the light source/TIA/ADC power consumption in the camera will be very large if working at such a high frequency. The incoming data fetched from memory will also be a bottleneck, which might not be able to fast enough to feed the 10 GHz optical core. In Line 297, the memory part is completely ignored when compared with digital computers, which might not be a fair comparison even in the near future. Only multiplications are done in optics, the partial product summation is done digitally, especially when it requires 4-pass, which raises concerns about the benefit of this SLM system for Transformer acceleration. More discussion on the system performance/efficiency is recommended.
7.	The paper title optical Transformer is very broad, however the current paper only focuses on SLM-based free-space optics, which suffer from bulky optical setups (low compactness) and noise/alignment/sensitivity issues in practical deployment. More discussion and comparison on other integrated photonic/diffractive hardware is important and necessary. Otherwise, the scope/title of the paper is more suitable to be narrowed down to an SLM-based Transformer acceleration platform.


**Questions:**

My questions are listed in the above weaknesses.

**Limitations:**

Yes.

---

> ### Author Rebuttal · Authors · 2023-08-09
>
> **Re: Novelty and Contributions to ML**
>
> We are not aware of any previous study on how ONNs would behave in the regime of LLMs, which are at least 100 − 1000× larger than any model simulated for ONN hardware so far. Due to the unavoidable noise in analog physical computing, the fact that ONNs work well for small-scale ML models (mostly developed for computer vision) does not imply they would work equally well for much larger ML models made for language processing. This work contributes to our understanding of whether current developments in LLMs could be suitable (and what the consequences would be) in the context of optical hardware, and vice-versa.
>
> **Re: LLM Inference Caching, Weights-in-Place vs Streaming Weights**
>
> In the case of attention, one might imagine only updating the in-place k/v data incrementally with new tokens and computing attention heads one-at-a-time. This way, the data re-use for attention operations is recovered. This kind of incremental writing and re-use may give the advantage to weight-stationary systems over streaming weights, where they would need to be reloaded. But it does require a lot of weight "memory".
>
> We note that caching has its own memory issues --- it uses an enormous amount of memory which in LLMs may require offloading to off-chip memory (which is an incredibly expensive data-access overhead [1]). These issues with large caches also suggest that a naively implemented LLM on ONN platforms can be thought of as a way to save memory rather than energy: In a scenario where caching is the default, one might imagine replacing it with a fixed-size, fast accelerator system that can just recompute the data nearly for free. In general, and even for GPU, compute is cheap while data access is expensive.
>
> Many Transformer architectures can or must perform the full attention/MLP computations in parallel. Some examples include: vision [2], language [3], and transfer learning on downstream tasks. We are interested in Transformers asmulti-purpose models that achieve state-of-the-art performance in tasks beyond language generation.
>
> **Re: The Amount of Optical vs Digital Operations**
>
> Most operations in LLMs are linear, and can be readily implemented by optics. In our experiment, only part of these linear operations were run optically on our experimental setup (marked with the laser icon in main text Fig. 2) for the purpose of error characterization. In our simulation, all linear operations were simulated as running optically with the experimentally derived error/noise model described above. We apologize for any confusion this might have caused. We only subsampled layers for the experiment because the available hardware we had for our prototype system ran at limited speeds.
>
> **Re: Shot Noise vs Other System Variations**
>
> As for our studies of energy scaling, it was necessary to consider shot noise independently from systematic error, because, in the low-light regime, shot noise eventually dominates over other sources of noise or error.
>
> **Re: Suitability of Gaussian Error Profile**
>
> We chose the Gaussian error profile because it very closely matched our experimental measurements (see Section 4/Fig. 3 for results, Appendicies B, C for further discussion of the experimental procedure, data sampling, and analysis of precision). That is, the Gaussian profile was observed, not assumed.
>
> **Re: Efficiency and Feasibility at High Speeds**
>
> Our energy estimates already account for components (ADC, DAC, modulators, etc.) running at the 10 GHz speed, which can be readily achieved by using integrated electro-optic modulators. However, in weight-stationary systems, only inputs need to be updated at this speed, but the weights need to be updated significantly less often if at all.
>
> We acknowledged in the discussion section that a whole system to achieve real speedup/efficiency advantages has yet to be realized. This is still an open challenge, and figuring out how to supply enough data to run at those speeds is definitely another big challenge.
>
> **Re: Data Access Cost (line 297 main text)**
>
> The statement made on line 297 applies to the loading weight only; all other memory costs are considered. The energy cost for one-time loading of weights can be ignored as long as each loading is sufficiently reused by working on large batches.
>
> **Re: Cost of Summation**
>
> Summing the multiplication results happens as the light is fanned in to produce the final magnitude of current corresponding to the value of the dot product, so the operation is not performed digitally, and happens as part of detection.
>
> The four-pass method for dealing with non-negative values in the setup does introduce extra data access costs, but they are insignificant for energy scaling. This is because the four-pass summation happens only once for each dot product, regardless of vector dimension, while the number of optical multiply-and-accumulate scales with vector dimension. Also, ONN systems using coherent light avoid this entirely.
>
> **Re: Discussion on Integrated Photonic Platforms**
>
> We agree with the referee that the optical energy scaling law in this work should be discussed in the context of other experimental platforms, since they are all promising contenders for achieving an optical advantage. We plan to revise our manuscript by adding discussion according to the outline provided in the section of **Comparison to Other Experimental Platforms** in the general rebuttal.
>
> [1] Pope et al. Efficiently Scaling Transformer Inference. arXiv:2211.05102. (2022)
>
> [2] Dosovitskiy et al. An image is worth 16x16 words: Transformers for image recognition at scale. ICLR. (2021)
>
> [3] Devlin et al. BERT: Pre-training of deep bidirectional transformers for language understanding. NAACL. (2019)

---

> > ### Comment · Reviewer_5GP2 · 2023-08-17
> > **Further Comments**
> >
> > Thanks for the responses.
> > 1. LLM Inference Caching, Weights-in-Place vs Streaming Weights
> > Even though the author claims a 10GHz SLM, for most weight-in-place integrated photonic accelerators to have low power, lowloss and small area, the switching speed is much lower than the computing speed (1 us vs 100 ps). It is not justified which workload, even for the referred Transformer that performs the full attention/MLP computations, can amortize such a high reprogramming cost. I don't think the batch/token for any vision/language task can merge this gap. So the claimed advantage of weight-in-place design over streaming one is not convincing.
> > 2.  Shot Noise vs Other System Variations.
> > The claim is for free-space optics (some prior work indeed claims low light, like sub-photon per MAC), but not for current integrated photonics. Given the current PD sensitivity, responsivity, and OSNR limit, it is not convincing why shot-noise will dominate. More discussion is needed if this paper is for general optical architectures.
> > 3. Data Access Cost (line 297 main text)
> > Similar to the above comments, reusing itself is hard to amortize the memory loading and device programming latency for most optical hardware, even for large batches. Some special computer architecture designs are necessary to hide the latency, which should be pointed out. This is the fundamental bottleneck for most analog hardware accelerators. It definitely requires cross-layer solutions to mitigate it, not just reusing..
> > 4. Cost of Summation
> > The four-pass method will introduce ~4x the latency and energy cost compared to one-shot computing. I still don't quite understand why this is insignificant in the overall efficiency/performance.
> > 5. Discussion on Integrated Photonic Platforms
> > This is the critical limitation of this paper, which claims "optical transformer" but lacks systematic analysis/comparison across different optical hardware platforms, and it requires significant modification/major revision of the paper.

---

> > > ### Author Response · Authors · 2023-08-21
> > > **Re: Further Comments**
> > >
> > > We thank the reviewer again for the additional feedback.
> > >
> > > **Amortized Loading Costs**
> > >
> > > For a fully weights-in-place system, the weights do not need to be switched. In other cases, Transformers' weight reuse can be sufficient to amortize loading costs. Typically, recent LLMs have sequence length $L \sim 10^4$. Without batching, at 10-GHz input, this only requires ~MHz-regime switching of weights, ~1us latency. For energy, we discussed this scenario in appendix G. For example, with 10G weights memory, ONNs enjoy a > 100x energy-efficiency advantage for the largest models, with models smaller than FUTURE-129T retaining nearly their full advantage.
> > >
> > > For attention weights-in-place may not be worthwhile as frequent switching is necessary. In all of our calculations we already assume that attention operations are computed with a streaming-weights approach. In this case, the weights are streamed by a light modulator in a similar fashion as the input data. We also acknowledge this at the end of section 4 (line 313). Consequently, implementing attention efficiently is hard, but MLP is the majority of Transformers' compute; ONNs still achieve an advantage in total MACs versus total energy (Appendix E).
> > >
> > > **Clarification About Noise/Error Analysis**
> > >
> > > We did not wish to claim that shot noise may dominate in most ONN systems, but rather to point out that a certain number of photons is necessary for a particular SNR requirement. Testing this requires examining the shot-noise-dominant regime. There also have been ONNs using low light, <100 photons for each photodetection [2, 4]. This translates to ~10 SNR, which is greater than hardware error.
> > >
> > > **Latency**
> > >
> > > Proposing a detailed specialized computer architecture is beyond the scope of our work. We acknowledge that the design of a suitably fast and efficient ONN system is still an open question. We hope that future ONNs continue to mature in this direction.
> > >
> > > **Cost of Summation**
> > >
> > > Altering energy costs by small constant factors (<10x) does not affect the main conclusions of this work:
> > >
> > > - **At the large scale, ONNs could achieve orders-of-magnitude energy-efficiency advantages over GPU**. As long as the overhead cost is also not orders-of-magnitude more expensive, it will not affect this claim
> > > - **The ONN advantage scales with the model size**. An increase in data-access cost only shifts the still asymptotically different energy curve
> > > - **The energy calculations are estimates**. We intended for our energy calculations to offer a discussion about ONNs' efficiency, not to predict specific numbers exactly; The discussions about scale and energy use of Transformers would not change
> > >
> > > Also, we remark that there would be *less than a 4x penalty* --- there are other considerations that would be unaffected but were a significant portion of the costs of digital ops. The presence of activations like ReLU and softmax means only two passes are needed in certain scenarios. Coherent-light ONNs can natively process real numbers (see systems comparison in Author Rebuttal).
> > >
> > >
> > > **Comparison to Other Platforms**
> > >
> > > Our work concludes that Transformers *can* run and are worth running in the presence of common analog-optical-system behaviors and pathologies. This suggests that in general, the use of optics to accelerate these large-scale models is worth pursuing. These central claims stand regardless of our limited discussion about the specifics of other ONN systems. In pursuit of these higher-level claims, we considered aspects that are fairly general:
> > >
> > > - Optical fan-out/in and related energy advantages are common to many ONNs
> > > - Other ONNs do have similar error profiles to ours [1, 2]
> > > - The scale-relative behavior (rescaling of operands) is common in analog computing
> > > - We emulated imprecision via the use of real LUTs
> > > - We tested Transformers at different precision levels, not just those in experiment (Appendix B.4, main text Fig. 3)
> > > - We assumed the use of memory, DAC, etc., which ONN accelerators require
> > > - We deliberately used commonplace techniques to make conclusions that do *not* rely on the details of a bespoke software or hardware approach.
> > > - Our energy estimations and assumptions are similar to the approach used for other ONN architectures in the literature [3, 4]
> > >
> > > These encompass much of the designs of ONNs, sufficient to paint the general picture of ONNs and Transformers. Nevertheless, we agree that this discussion would be helpful to include in the article.
> > >
> > > [1] Feldmann et al. Parallel convolution processing using an integrated photonic tensor core. arXiv:2002.00281. (2020)
> > >
> > > [2] Sludds et al. Delocalized photonic deep learning on the internet's edge. Science (Vol. 378, Issue 6617, pp. 270–276). (2022)
> > >
> > > [3] Hamerly et al. Large-scale optical neural networks based on photoelectric multiplication. Physical Review X, 9(2):021032. (2019)
> > >
> > > [4] Wang et al. An optical neural network using less than 1 photon per multiplication. Nature Communications, 13 (1). (2022)

---

### Official Review · Reviewer_F5Gn · 2023-06-26

**Soundness:** 3 good
**Presentation:** 2 fair
**Contribution:** 3 good
**Rating:** 5
**Confidence:** 4

**Summary:**

This paper propose a photonic hardware accelerator to process the inferences of large language models, i.e., transformers, using optical multiply-accumulate (MAC) operations. Optical MACs are suitable for computations with large operands, thereby leading to asymptotic energy advantages over the digital hardware accelerators.




**Strengths:**

1. The paper is well-organized.
2. The paper works on an important problem.

**Weaknesses:**

1. The paper does NOT consider the energy consumption of analog-to-digital converters, digital-to-analog converters, and various memories such as on-chip SRAM and off-chip DRAM. I totally agree with the cornstone of this paper, which is optical MACs or matrix multiplications are super energy-efficient. However, gaining this energy advantage is not easy. Reading 530B parameters of a transformer, converting these many digital parameters to analog optical signials, and converting the analog optical result signials back to digital values may donimate the energy consumption of an inference. As a result the energy efficiency improvement may not be very large.

Please check the comparison in this paper:
W. Liu, W. Liu, Y. Ye, Q. Lou, Y. Xie and L. Jiang, "HolyLight: A Nanophotonic Accelerator for Deep Learning in Data Centers," 2019 Design, Automation & Test in Europe Conference & Exhibition (DATE), Florence, Italy, 2019, pp. 1483-1488, doi: 10.23919/DATE.2019.8715195.

**Questions:**

Please comment on the energy consumption of analog-to-digital converters, digital-to-analog converters, and various memories such as on-chip SRAM and off-chip DRAM during the otpical transformer inferences.

**Limitations:**

No potential negative societal impact.

---

> ### Author Rebuttal · Authors · 2023-08-09
>
> Hello and thank you for your feedback. The overheads related to data access (RAM, DAC/ADC) are indeed very important in considering whether an ONN platform may have any energy advantage. All energy values reported in this work did take those into account. We acknowledge that explicitly mentioning in the main text the details of what costs are included would have been more clear.
>
> These data access costs are also important to consider because they shed light on how optics-based platforms work differently from digital ones. We explained our approach in Section 2 (see line 107) and summarize it as follows:
>
> - Using the effective model bit precision we found in a small ablation study (Appendix C), we estimated the per-use cost of DAC, ADC, modulators, TIA, DRAM, and SRAM at these precisions. We assumed energy quantities found from existing products' datasheets or reported in other research (Appendix D).
> - We used these energy quantities to compute the total energy cost of all data access in the models (Figure 2, appendicies D, E, F). Each access of a single tensor element counts as a single use of the relevant DAC/ADC/memory.
> - We showed that even with ADC/DAC and other data-access related costs accounted for, there is still a significant and scaling energy advantage for larger models, but that the overhead makes running smaller ones optically less worthwhile (Section 4.3).
> - We broke down the estimated energy costs for Transformer models to see the contribution of each component (Appendix E), and indeed found that memory access, DAC, and ADC are the overwhelming majority of energy costs in our estimates.
> - We provided our code for producing our estimates, where the user can reproduce the values calculated and change the energy quantities.
>
> If these data access overheads were correctly accounted for and are very expensive, then a reasonable question to ask is how ONNs obtain such large energy advantages when running Transformers. A focus of our work is to highlight the idea that even if these data-access costs are expensive, they may be amortized by re-use of data in the optical domain. This has been investigated in previous works [1. 2], but we are aware of no existing study that discusses how this is affected by model architecture, what model scales the asymptotic advantage of optics overcomes these additional overhead versus digital systems, and what makes existing popular architectures (such as Transformers) well or not-well suited. As an example, consider a matrix-matrix product with operands of shape ($m \times n$), ($n \times d$). The total number of MACs is $mnd$. This operation requires loading of ($mn + nd$) elements, and storage (ADC+memory, $E_\mathrm{store}$) of the resulting matrix's $md$ elements. Each element of the loaded matrices is reused. In a weights-in-place system, the cost for loading the $nd$ elements is ignored, but all other calculations are identical. The rows of the ($m \times n$) matrix each get fanned out optically (free) to create $d$ copies, and each column of the ($n \times d$) matrix gets reused $m$ times. So the total cost of the data access (and we address optical energy scaling in Section 4.2) is $E_\mathrm{load}(mn + nd) + E_\mathrm{store}(md)$. Unlike for digital computers, this is not proportional to the number of MACs, and therefore results in an asymptotic advantage: the energy per MAC is $O(\frac{1}{m} + \frac{1}{n} + \frac{1}{d})$ [1, 2]. It follows that models with large weight/activation matrices would be best suited for achieving an optical advantage, hence our interest in Transformers, their large MLP blocks, and their parallel-processing of many tokens with the same weights. We discussed how design decisions like these are critical in creating DNN architectures that can be run on an ONN advantageously (Section 5).
>
> We thank the reviewer for bringing to our attention [3] --- we believe that the prospect of designing ONNs without ADCs entirely is exciting. The energy usage for the ADC (in Table 1 of the article) also appears to be roughly in agreement with our estimate: assuming 8 bits of precision, 2048 mW for 1024 uses at a time, and 1.28 GHz speed, the energy usage appears to be 1.56 pJ per 8-bit sample. Our estimate in this work was (appendix D) 3.17 pJ per 7-bit sample.
>
> We wish to reiterate that **all ONN energy cost estimates in this work did include the costs of DAC, ADC, and memory access**.
>
> [1] Hamerly et al. Large-scale optical neural networks based on photoelectric multiplication. Physical Review X, 9(2):021032. (2019)
>
> [2] Wang et al. An optical neural network using less than 1 photon per multiplication. Nature Communications, 13 (1). (2022)
>
> [3] Liu et al. HolyLight: A Nanophotonic Accelerator for Deep Learning in Data Centers. 2019 Design, Automation & Test in Europe Conference & Exhibition (DATE), Florence, Italy. (2019)

---

> > ### Comment · Reviewer_F5Gn · 2023-08-14
> >
> > Thanks for the rebuttal. I increase my score.

---

### Official Review · Reviewer_DVPz · 2023-07-12

**Soundness:** 3 good
**Presentation:** 3 good
**Contribution:** 3 good
**Rating:** 6
**Confidence:** 3

**Summary:**

This paper explores the feasibility and benefits of employing optical computing techniques for machine learning and specifically focusing on large language models (LLMs).  The paper builds upon earlier work on optical neural networks, primarily [61] (Wang et al., Nature, 2022), which experimentally demonstrated the feasibility of performing dot products optically in a two layer neural network applied to MNIST while achieving around 90% accuracy at about one photon per multiplication optical energy.  As LLM computations involved a large and rapidly growing number of multiply accumulate operations, the objective of the paper is to explore whether optical techniques can yield benefits over existing CMOS-based accelerators (GPUs, TPUs).   The paper tackles this by employing a simulation based methodology where the simulator attempts to model the various noise sources (systemic, shot noise) along with limited precision of a potential electro-optical system.  The evaluation shows that as LLMs continue to scale such optical systems may potentially yield many order of magnitude benefits in terms of energy efficiency over current approaches.

**Strengths:**

Makes a reasonably strong case for further exploration of optical computing for large language models.

**Weaknesses:**

Not enough discussion of the remaining challenges that need to be overcome to make such systems competitive in reality.

Somewhat limited contributions in-so-far as earlier works have already explored using optical for ML.

Unclear how accurate the simulation methodology is.

Some aspects unclear.

**Questions:**

How far are current optical systems from the 10GHz operating frequency assumed in this paper (Line 291) and what would Figure 5 look like and to what extent would the main conclusions about the benefits of optical be undermined if more modest (or perhaps realistic) rates were assumed?  How much improvement is required in this direction to achieve the hoped for benefits?  What frequency do current modulators operate at?

How do you separate out shot noise from systematic noise in the measurements in Figure 3?

Does the y-axis in Figure 4 of the main paper include the electrical energy overheads like those in Figure 3 and 4 in the supplemental? Figure 3 in the supplemental seems to show optical energy is entirely negligible so in this case the accuracy of Figure 4 in the main paper would be entirely dependent upon a full accounting of all other energy sources.   What assurances can you give that all significant sources of non-optical energy are properly accounted for?

The assumption of weights-in-place seems quite unrealistic and while the supplemental discusses (Section G) a "chunking" technique to scale up to larger numbers of parameters it was unclear whether the assumptions used to plot Figure 6 in the supplemental section are optimistic or conservative and to what extent.  Please comment.

Transformers are having a lot of impact, which makes the focus of this paper on them make sense, but can you comment on how much optical may other network architectures?   Accelerators like GPUs and TPUs try to be relatively general purpose and so were able to be quickly retargeted to transformer based models without needing much in the way of hardware changes.  Can your simulation based study approach yield insights about whether optical will help networks other than transformers and/or under which circumstances?

How accurate is the simulation methodology in Figure 2?   Can you attempt to compare your simulators predictions versus your actual hardware for a small network like the one studied in [61]?

I did not follow what the lookup-table (LUT) in the simulator does.  What values do the LUTs in the simulator contain and what quantity is used to index into the LUT?

**Limitations:**

There is some discussion of limitations.  If the paper is meant to rally others to work on optical techniques for machine learning it would be helpful if the authors could more systematically highlight the remaining challenges to making such systems practical.

---

> ### Author Rebuttal · Authors · 2023-08-09
>
> **Re: "How far are current optical systems from the 10GHz operating frequency", "...benefits of optical be undermined if more modest rates were assumed?**
>
> If the system were to be run at lower frequency the energy estimates would not change significantly for smaller models, but can for the hypothetical future models. The main concern about the relationship between speed and energy cost is the cost of maintaining the weights in a weight-stationary system. In our estimates we modeled this after the power consumption of a simple LCD display (Appendix D). If the frequency is slower, then the amount of energy expended to run per processed sample would therefore be higher. For example, for FUTURE-4q, the largest hypothetical future model, when we change update rate from 10 GHz to 1 GHz, the energy-advantage factor would change from ~8000x to ~6000x (can be reproduced by simply multiplying the weight maintenance cost value by 10 in provided source code). For smaller models the difference is negligible. Also, at lower speeds ADC/DAC and other components may become significantly cheaper, which we did not account for in this calculation. Our estimate for the weight maintenance cost is quite conservative and based on existing components, not those that future models would be run on.
>
> 10 GHz is already achieved by transceivers in telecom applications. Other components can also run more quickly:
> - DAC/ADC can exceed 10 GHz [2]
> - Current modulators can run in the regime of 100 GHz [3]
> - 30 GHz Si-Ge photodiodes have been demonstrated in ONNs [4]
>
> We wish to clarify that only the input data needs to be modulated at 10 GHz. Weight-stationary systems do not need to be updated at a high speed or at all.
>
> **Re: limitations of weights-in-place**
> The scalability of the weights-in-place approach is indeed a concern, but the requirement of more GPUs for more VRAM is a concern too when weights/activations become large. Hence the approach in appendix G, which is conservative because GPU-GPU communication is estimated as only being as expensive as DRAM. We remark that Fig. 6 is mostly related to the cost of input reloading for multiple chunks in cases where the processing cannot be parallelized across devices. All comparisons were against a single, ``ideal'' GPU (300 fJ/MAC).
>
> We wish to point out that even in streaming-weight systems there can still be significant energy advantage from data re-use, so the foundational principle of the energy-advantage argument for optics still applies: when the weights are streamed, they may still re-used by a factor of the batch size times the number of vectors in the input token sequence to be processed.
>
>
> **Re: "somewhat limited contributions"**
>
> While previous studies have shown ONNs running small-scale computer vision tasks such as MNIST, this is the first to show ONNs' performance on reasonably large models that include optical shot noise and errors. This is important because the energy advantages for ONNs only become large at enormous scales. We wish to clarify here what concrete findings and concepts we believe are valuable:
>
> - We presented a simple method and techniques that proved that a Transformer model at sufficient scale to be used in practical tasks *can* run on an ONN accelerator despite real-world hardware errors and shot noise. This is a new, concrete piece of information that could not be inferred from previous works on ONNs that considered smaller models with different architectures. The largest such *simulated* model we are aware of is AlexNet [1]
> - We documented how scaling of models is related to energy usage.
> - That Transformers achieve the efficient scaling (Fig. 4) is a nontrivial result --- The scaling is highly dependent on model statistics. Thus, our findings were not trivial to assume a priori just because previous ONN literature has investigated the effect in other architectures,  tasks, etc.
> - Our energy estimates provide a perspective on what might be achievable on future ONN hardware if their creation is to be pursued. ONNs are still an emerging technology, so there was an open question of whether further development should be pursued at scale.
> - We provide a list of specifications for what a theoretical accelerator might need to accomplish based on our findings.
>
> **Re: Separating Shot Noise From Systematic Errors in Experiment**
>
> We define systematic errors as corrupted output due to deficiencies intrinsic to the hardware. Thus, it is identical across runs of an experiment. We use this among other approaches to isolate them:
>
> - We use high photon counts in our experiments. This directly reduces shot noise.
> - We average the results of 10 trials in experiments. Because of how systematic error works, it will remain persistent. This also eliminates noise from other sources than shot noise.
> - We fit a calibration curve to the averaged results. Any deviation is systematic error.
>
> **Re: Viability of other networks**
>
> While we did not simulate other models, we do not believe that all architectures would be as suitable for ONNs. We chose Transformers because if ONNs were only good at running one model, the model should be useful for many different tasks. We emphasize that the purpose of our simulations was to emulate the behavior of the actual hardware, and so simulating other models is possible. Their performance is a separate issue.
>
> [1] Hamerly et al. Large-scale optical neural networks based on photoelectric multiplication. Physical Review X, 9(2):021032. (2019)
>
> [2] Liu et al. A 10GS/s 8b 25fJ/c-s 2850um2 Two-Step time-Domain ADC using delay-tracking Pipelined-SAR TDC with 500fs time step in 14nm CMOS technology, IEEE (ISSCC). (2022)
>
> [3] Wang et al. Achieving beyond-100-GHz large-signal modulation bandwidth in hybrid silicon photonics Mach Zehnder modulators using thin film lithium niobate. APL Photonics; 4 (9): 096101. (2019)
>
> [4] Ashtiani et al. An on-chip photonic deep neural network for image classification. Nature 606, 501–506. (2022)

---

### Official Review · Reviewer_32hz · 2023-07-14

**Soundness:** 3 good
**Presentation:** 4 excellent
**Contribution:** 3 good
**Rating:** 8
**Confidence:** 3

**Summary:**

The authors perform experimental analysis with a spatial light modulator to optically perform the computations of the linear components of the Transformer architecture. These measurements allow them to create a noise-model that is then used to simulate a GPT-2 like model and measure performance (validation perplexity) in function of model parameters (system noise, not including optical shot noise). These optical systems are physically constrained by "optical shot noise" that dictates the minimum number of photons to achieve a target precision. This optical shot noise scales favorably with larger models, and the authors then extrapolate their observations to existing large Transformer models (like PaLM), and further to even larger hypothetical future models, showing a substantial advantage of a large scale hypothetical optical system over existing electrical systems (the energy advantage scaling with the width of the model).

**Strengths:**

1. While there have been previous works that examine smaller scale optical neural networks and the optical shot noise scaling behavior seems to be well established (I'm inferring this from the works cited in the submission, I'm not familiar with the area), the authors highlight the importance of the scaling behavior in context of the Transformar architectures that are typically used with large language models. I think the combination of innovative research in the field of optical computation with the computation requirements of applied machine learning models at the largest scales is particularly interesting.

2. The authors used small scale experiments to establish realistic system properties and then simulated an entire Transformer to check how the physical properties would influence overall (validation) performance of the model. They did this at a scale (GPT-2 like quantized model with 15M-416M parameters) that seems large enough to gather realistic predictions.

3. The interpolation of the simulated data to much larger models is necessarily based on many assumptions, both in scaling from the simulation to larger models, and with respect to the hypothetical hardware that would run very large models on optical hardware. The appendix goes into some detail on the different assumptions that led to the conclusions summarized in the main part of the article.

4. The overall presentation of the work seems adequate for a public that is knowledgeable in machine learning, but probably has much less experience with physical properties of optical computational hardware, which is no easy task given the relatively large gap between these domains.


**Weaknesses:**

1. All the discussion of the optical vs. electrical implementation of the computation is in the lens of energy consumption. After reading the paper I'm not sure how the proposed architecture would fare with respect to latency/speed, or other constraints (e.g. if there is a theoretical limitation to the size – or cost – of different components that scales very differently between the currently used IC technology of purely digital microchips vs. optical components). Latency is mentioned in Appendix G, but there is no mention of these other dimensions in the main text.

2. The authors highlight the scaling behavior of optical shot noise, but other sources of noise (called "systematic errors" in the paper) are simply measured for the system at hand, and no information is given how these other sources of noise would scale when the system is scaled. For example, I assume that any realistic scaling would require miniaturization of the optical components, and I would assume that this miniaturization also causes the systematic errors to change in relative magnitude. It would be interesting to at least briefly discuss the scaling behavior of these other sources of noise.


**Questions:**

Small things I noticed while reading the paper:

a) 155-157: not clear what this means: “We collected lookup tables (LUTs)—mappings of the available discrete levels in both the display and SLM devices—and used them to train a “LUT-aware” optical Transformer model to run on the setup.” why are the LUTs needed? what does “collected” refer to? (measurements used for downstream analysis? used for the computation itself?) … maybe referencing what is exactly meant with LUTs (is it a hardware component?) would also help here

b) 171, 179: the abbreviation QAT is used without being introduced (it’s spelled out on line 194)

c) Table 1 is a bit hard to read because the “Setting” column spans multiple rows, but the text neatly aligns (Hardware->QAT, Simulation->Eval, etc) with the individual rows (one could e.g. add more horizontal lines in columns 2+ to make this clearer)

d) 196: when saying “int8” is it actually “uint8” ?

e) Figure 5 I find it surprising to see TPU int8 use 800fJ/MAC vs. NVIDIA A1000 fp16 300 fJ/MAC – I could not readily find these numbers in [47] – how exactly were they computed?

f) 329: either “study” or “illustrate” (but not “our studies illustrates”)

g) Appendix, Figure 4 (left): “Digital Ops” are not visible – why? If they are zero (e.g. “not doing any compute”) then that should be mentioned.

h) Appendix, Figure 4 (right): the colors are hard to tell apart (especially it would be interesting to clearly see optical vs. electrical). It’s also not very clear what are “Ele FF” and “Ele Attn FF” (maybe ReLU6, LayerNorm, and Add from Figure 2?)


**Limitations:**

I don't see a negative societal impact of the presented work, other than making Transformers more energy efficient would allow to scale them further than otherwise possible, and accordingly accentuate any potential opportunities and risks of models such as LLMs. I don't think it's required to point these out explicitly in the paper.

There seem to be a number of technical limitations with respect to the predictions in the sense that there is a lot of uncertainty whether the hardware to run optical neural networks could be scaled up as much as state of the art digital circuits. But the authors make it clear already in the abstract that their main interest is in establishing a scaling law and not concrete predictions about the future of the implementation of large scale optical neural networks.

---

> ### Author Rebuttal · Authors · 2023-08-09
>
> We are grateful for the reviewer's helpful comments. We provide here further explanation on the reviewer's questions and concerns in a point-by-point manner.
>
> **Re: "All the discussion ... in the lens of energy consumption"**
>
> Latency and speed are important factors in determining the viability of ONN platforms, and we agree that not providing in-depth analysis is a limitation of our work. One important reason why we chose to focus on the energy efficiency of ONNs is that the energy consumption of different ONN architectures can be analyzed in a universal framework based on the common physical principles they share (such as data re-use, optical transmission advantage, and photon detection noise). Speed and latency, on the other hand, can vary drastically from architecture to architecture and many ONNs are optimized towards particular operations such as convolutions [1] or for black-box reservoir [2] or physical-neural-network computing [3] where the equivalent number of "flops'' is likely high but undefined/incalculable.
>
> The throughput advantage is often related to optics' high bandwidth and parallelism. Presently, though, ONNs are still an emerging technology and many implementations, such as ours, are proofs-of-principle that have not been optimized for speed. But we do not see any fundamental limitation for ONNs to update at 10 GHz speeds or higher, and ONN technology itself is progressing, with promising demonstrations at significant speeds [4].
>
> **Re: sources of error only measured for the system at hand**
>
> We acknowledge that the systematic error analysis we reported in this work is specific to our system and not entirely generalizable to ONNs as a whole. That said, the systematic error of our experimental setup is typical among ONNs [4, 5], and therefore is representative of the numerical precision that can be achieved by macroscopic or integrated systems in many cases. It is also possible to reduce systematic error via more sophisticated calibration techniques. By definition, systematic errors are consistent defects that exhibit the exact same behavior from run to run. Thus, some ONN platforms, such as Mach-Zehnder Interferometer (MZI) arrays [7], are designed with more configurable components and systematic errors can be eliminated by a calibration procedure that accounts for device defects.
>
> **Re: "how these other sources of noise would scale"**
>
> While we agree that miniaturization of ONN devices *may* lead to higher systematic error or device-to-device variation, we think the continued scaling of ONNs do not hinge on the miniaturization. We believe that systematic error behavior has more to do with the particular ONN implementation than the scale. One could also imagine scaling up a system by fanning out data to multiple copies of the same system, meaning that the error would be the same at worst.
>
> **Re: "Why are the LUTs needed? what does “collected” refer to?"**
>
> The LUTs, which we collected for the OLED display and SLM, are mappings from indices of the discrete values supported by the devices' displays to actual measured intensity values in the lab. They are collected by measuring the levels of transmitted light for each possible display (with SLM at full transmission) and SLM setting (with display fixed). These differ from simple quantization in that the intensity values are not linearly scaled with the input index, and the smallest SLM weight value is 0.02 instead of 0. Thus by incorporating these measured LUTs directly in our simulation instead of a naive 8-bit quantization scheme, we allow both in QAT and during inference the model to function only with the hardware's capabilities/deficiencies. Backpropagation is carried out using the straight-through estimator, but unlike QAT once the rounding operation produces the quantized int8 activations, they directly index the LUTs to produce the floating-point activations instead of dequantizing.
>
> **Re: "...TPU int8 use 800fJ/MAC... how were they computed?"**
>
> For the GPU, we used the peak performance numbers specified by NVIDIA for the device, as well as the TDP to estimate power. For TPU, the comparison was made against the older TPUv1. This is easier to discern in Fig. 3 of the revised version of that article [7] where the int8 TPUv1 is shown as achieving roughly $9 * 10^4$ GOps/s at roughly 70W of peak power, ~778 fJ/Op.
>
> **Re: Digital ops not visible**
>
> The digital operations are present in the chart in Fig. 4, but we recognize that it is difficult to see. For all but the smallest models, there are digital operations but the fraction is so small that it is effectively zero.
>
> The "Ele *'' ops refer to the electrical overhead costs of a particular part of the model. For example, "Ele Attn QKT Ld'' refers to the **electronics** costs of **loading** the operands for the part of the attention operation where the $Q$ and $K$ matrices are loaded to perform the product $QK^T$. "Attn FF'' refers to the linear layers in the attention part of the Transformer where the $Q$, $K$, and $V$ tensors are derived. It also includes the final linear mapping at the end of attention. The other "FF'' labels refer to the layers in the MLP blocks.
>
> [1] Xu et al. 11 TOPS photonic convolutional accelerator for optical neural networks. Nature, 589(7840):44–51, (2021)
>
> [2] Lupo et al. Fully Analog Photonic Deep Reservoir Computer Based on Frequency Multiplexing.arXiv: 2305.08892 (2023)
>
> [3] Wright et al. Deep physical neural networks trained with backpropagation. Nature 601, 549–555 (2022)
>
> [4] Feldmann et al. Parallel convolution processing using an integrated photonic tensor core. arXiv preprint arXiv:2002.00281 (2020)
>
> [5] W. Zhang et al. Silicon microring synapses enable photonic deep learning beyond 9-bit precision. Optica  9, 579-584 (2022)
>
> [6] Shen et al. Deep learning with coherent nanophotonic circuits. Nature Photonics, 11(7):441, (2017)
>
> [7] Reuther et al. AI and ML Accelerator Survey and Trends. 2022 IEEE (HPEC) (2022)

---

> > ### Comment · Reviewer_32hz · 2023-08-16
> >
> > I would like to thank the authors for their excellent rebuttal, both in the common part, as well as in the individual follow-ups.
> >
> > This has reinforced the confidence in my rating, and it will be interesting to see if other reviewers adapt their scores accordingly.
> >
> > Some follow-up comments from my side:
> >
> > 1. I'm looking forward to the inclusion of additional optical systems in the paper ("Comparison to Other Experimental Platforms" from the common rebuttal). In particular, this would give a good opportunity to include some of the discussion above about speed/latency, errors from other systems and how they likely scale with miniaturization. I'm somewhat skeptical of the authors' claim "One could also imagine scaling up a system by fanning out data to multiple copies of the same system", as this only allows for very limited scaling.
> >
> > 2. Looking at Figure 4 in (Reuther, 2022) still leaves me wondering why Figure 5 in the submission has the efficiency ordering ASIC (100 fJ/MAC) > A100 (300 fJ/MAC) > TPUv1 (800 fJ/MAC), since that same Figure 4 in (Reuther, 2022) shows (approximately) 400W for 3e5 GOps/sec (which would be 1333 fJ/Op, and not 300 fJ/Op, and reverse the ordering A100/TPUv1).
> >
> > 3. I found the additional explanations as to the specific use of LUTs very insightful. I would appreciate it if the authors updated the main text in the submission to make this point clearer.

---

> > > ### Author Response · Authors · 2023-08-21
> > >
> > > We thank the reviewer again for their additional feedback.
> > >
> > > Regarding the GPU estimate, a general approach we took throughout this work was in cases of ambiguity we preferred to overestimate the GPU's efficiency and underestimate the ONN's advantages. We therefore aimed to provide as generous an estimate as possible for the NVIDIA A100 because finding consistent information about its real world, FP16, inference-mode performance was hard. For example the value reported by Reuther et. al appeared to be for training, which is more data-bottlenecked than inference. Also, the 400W GPU appears to be the SXM variant of the A100 which consumes an additional 100W of power. So to build an estimate we divided the rated power consumption of a typical GPU by the peak reported performance on NVIDIA's datasheet for the card; we assumed typical GPU power consumption of 200W for ~624 TFLOPs of compute. In the best case (200W, but a more typical power consumption for A100 would be ~250-300W), this yields roughly 300 fJ/MAC (depends on how MACs/FLOPs are counted, however). This way, despite ambiguous information about the A100's real-world characteristics, we could be reasonably confident that our reported ONN energy advantages were not overestimated. We agree with the reviewer that in practice the cost could be higher than estimated for GPU.
> > >
> > > We also agree that the work could benefit from additional discussion about related ONN works/hardware, miniaturization/scaling, and the LUTs. We hope to add these to a revised version if possible.

---

### Author Rebuttal · Authors · 2023-08-09

We wish to thank the reviewers for their time and dedication in providing valuable feedback for this work. We hope that the additional explanations we provided here serve to clarify the scope of this work and address some of the common concerns of the reviewers about our energy calculation assumptions and the applicability of our results to other ONN platforms. We have collected our responses to those concerns here so that they are available to everybody:

**Scope of This Work**

The scope of this work was to study what potential benefits could be had if future optics-based hardware were designed for accelerating large-scale neural networks. In essence, we aimed for results that could generalize well to the field of ONNs by selecting an existing popular DNN architecture and ONN platform. In this sense, while some reviewers have expressed their concern about the novelty of our ONN design, we wish to emphasize that we intentionally chose an already well-understood platform because of its many features that are common to ONNs in general. This is why we make claims about optical Transformers in general, and not our particular SLM-based system. We discussed broadly how they they could be achieved via ONN implementation. These specifications (Appendix G) are intended to be general - any platform capable of a certain amount of compute performance counts as a ``core'', any device that can emit or detect a scalar value is suitable for processing input/output vectors, and so on. These are a set of high-level requirements that we believe *any* ONN platform should target if aiming to achieve an energy-efficiency advantage, regardless of implementation details.

**Energy Calculations Always Include Electronics Overheads**

Several reviewers pointed out the importance of counting the overhead energy costs of loading/encoding/decoding/storing the data to be used in an optical accelerator. We apologize for any confusion, and we would like to clarify that **all energy estimates include the overhead costs of electronics for data access**. This includes digital-analog conversion, analog-digital conversion, memory read/write, modulation, weight maintenance costs, detection, and transimpedance amplification. These costs were a critical part of our analysis, since the energy advantage we discuss hinges on the abilities of optics to amortize these costs through data re-use and shot-noise-limited photon scaling. The details of our calculations, and all assumed values are in Appendices D, E, and F. We have also provided our source code for reproducing the calculations, where the user can change various energy quantities to experiment with how they affect the energy usage.

**Clarification and Motivation For Using Lookup Tables (LUTs)**

We included LUTs to model a kind of hardware error that is common to many optoelectronic devices. There are differences between the precision limitations of real devices and linearly-spaced quantization schemes often used for DNNs. The LUTs were collected for the organic LED display and spatial light modulators (SLMs). While these devices are commonly controlled by digital signals with evenly spaced discrete levels, the resultant output of these devices tends to be unevenly spaced because of their intrinsic nonlinear response or finite extinction ratios.

We incorporate these LUTs into both training and simulation. Backpropagation is carried out using the straight-through estimator just as for QAT, but unlike QAT once the rounding operation produces the quantized uint8 representations, the numbers are directly used to index the LUTs to produce the activations instead of dequantizing.

**Comparison to Other Experimental Platforms**

We recognize that further discussion about other platforms and how their similarities/differences to ours would provide useful background information. We plan to revise our manuscript by discussing how all these platforms are good contenders for optical LLMs, as long as they possess specific properties, such as optical data re-use, to support the optical scaling law. A summary of representative works is as follows:

- Wavelength-division-multiplexed Modulator Array [1, 2, 3]: Data is fed into a grid-like structure with resonators or phase-change materials to modulate the light field according to weights.
- Mach-Zehnder Interferometer (MZI) meshes [4]: These devices use cascaded networks of MZIs (which store weights) to implement matrix-vector multiplication.
- EOM-based convolution engines [5]: These leverage EOMs' toeplitz-matrix coupling of modes in the synthetic frequency dimension to implement convolutions.
- Coherent, SLM-based free-space ONNs [6]: A scheme very similar to ours, but supports real-number data.
- Coherent, free-space diffractive ONNs [7]: A scheme that uses optical depth in 3D space to encode a large amount of parameters for ONN layers.

[1] Mesaritakis et al. Micro ring resonators as building blocks for an all-optical high-speed reservoir-computing bit-pattern-recognition system. J. Opt. Soc. Am. B, 30(11):3048–3055 (2013).

[2] Feldmann et al. Parallel convolutional processing using an integrated photonic tensor core. Nature 589:52–58 (2021)

[3] Tait et al. Microring weight banks. IEEE Journal of Selected Topics in Quantum Electronics 22.6: 312-325 (2016)

[4] Shen et al, Deep learning with coherent nanophotonic circuits. Nature Photonics 11(7):441 (2017)

[5] Fan et al. Multidimensional Convolution Operation with Synthetic Frequency Dimensions in Photonics. Physical Review Applied 18 (2022)

[6] Spall et al. A. Fully reconfigurable coherent optical vector–matrix multiplication. Optics Letters 45(20): 5752–5755 (2020)

[7] Lin et al. All-optical machine learning using diffractive deep neural networks. Science 361.6406: 1004-1008 (2018)

---

### Decision · Program_Chairs · 2023-09-21

**Decision:**

Reject

**Comment:**

This paper received mixed reviews. However, only reviewer 5GP2 is a world-leading expert in the paper's domain of optical accelerators, and they are unconvinced even after extensive discussion.

After further discussion between AC and SAC, we have decided to reject the paper. Besides the multiple concerns raised by 5GP2 which were unresolved, we further agree that NeurIPS is not the best venue for this work, and suggest working on improvements suggested by the reviewers and submitting to a better fitting venue in the optics community.